# A survey of knowledge and attitudes towards antibiotic use and resistance among teachers in the Republic of Kenya: Implications for using teachers in raising public awareness of rational antibiotic use in school communities

**Patrick M. Mutua**[1]*, **Joshua. Mutiso**[2], **Michael M. Gicheru**[2]

1 Department of Public Health, School of Health and Human Sciences, Pwani University, Kilifi, Kenya,
2 Department of Zoological Sciences, School of Pure and Applied Sciences, Kenyatta University, Nairobi, Kenya

* patmbuvi@gmail.com

**Data Availability Statement:** All relevant data are within the manuscript. I have attached the

## Abstract

### Background

Antimicrobial resistance is a significant public health threat. In Kenya, schools are targeted for public education to promote knowledge and attitudes towards the proper use of antimicrobials. However, there are limited studies that have investigated teachers' knowledge and attitudes on antibiotic use.

### Methods

We conducted an online survey from December 2023 to January 2024 which included 608 primary and secondary school teachers, representing a response rate of 93% of the estimated sample size of 653 teachers. Data on the respondents' antibiotic knowledge score (seventeen questions) and attitude score (eleven questions) were analysed using IBM SPSS (Version 27). A binary logistic regression model was applied to analyze predicators of adequate knowledge and attitude on antibiotics.

### Results

The average knowledge score for antibiotic use was 9.2 out of 17. Among the sampled teachers, 82% had not received public awareness information on proper antibiotic use. More than 86% of the respondents incorrectly answered that antibiotics are effective against colds. The average attitude score on antibiotic use was 6.1 out of 11. A quarter of the interviewees agreed that they gave family members antibiotics wherever they fell sick. Respondents aged 40–49 years and teaching in secondary school had higher odds for adequate knowledge in antibiotic use.

questionnaire that we used to carry out the study with the manuscript as advised. Kindly find the attachment in the file named manuscript.

**Funding:** The author(s) received no specific funding for this work.

**Competing interests:** The authors have declared that no competing interests exist.

## Conclusions

This study has identified significant knowledge and attitude gaps that need to be addressed by policy makers to ensure rational antibiotics use among teachers and in ensuring effective use of teachers in raising awareness in school communities for rational antibiotic use.

## Introduction

Antimicrobial resistance (AMR) is the ability of pathogens to resist drugs to which they were previously susceptible [1,2]. The increasing burden of AMR infections has been cited as a threat to public health globally and has been associated with prolonged hospital admissions, high morbidity and mortality rates, increased health care costs and negative impacts on other medical interventions, such as surgery, organ transplantation, and caesarian deliveries [3–5]. While AMR is a natural process, human factors such as overprescription of antimicrobials, self-medication, underdosing, medical prescription of antimicrobials without laboratory tests to identify causative agents of infections, and poor water and sanitation, have been identified as some of the leading causes of AMR [6–9]. Globally, 4.95 million deaths associated with AMR were recorded in 2019, including 1.27 million deaths attributable to AMR infections [10]. In the same year, Kenya recorded 37,300 deaths associated with AMR which included 8,500 deaths attributable to AMR infections [11]. There are more AMR deaths recorded in the country than deaths caused by neoplasms, enteric diseases, maternal and neonatal deaths, diabetes or kidney deaths [11]. A recent study revealed increased misuse of antimicrobials during the COVID-19 pandemic [12] raising concerns about the effectiveness of measures that have been put in place to counter AMR. Furthermore, estimates indicate that by 2050, AMR will cause an estimated 10 million deaths per year and, with the current situation where the least developed countries are disproportionately most affected by AMR, Africa is projected to record approximately 4.15 million deaths annually by 2050 [13]. School communities have been targeted for imparting positive knowledge on and attitudes towards antimicrobial use and resistance. In a study conducted in Ghana, for example, picture drawing was demonstrated to be an effective intervention for instilling adequate knowledge on antimicrobial use and resistance among school going children [14]. In 2017, Kenya established National Action Plan (NAP) for the prevention and containment of AMR on the basis of the World Health Organization (WHO) guidelines for establishment of Global Action Plans on the Prevention and Containment of AMR [15]. The Kenya NAP comprises a National Antimicrobial Stewardship Interagency Committee (NASIC) and forty-seven County Antimicrobial Stewardship Interagency Committees (CASIC) [16]. In their reviews, the NASIC and CASIC identified the need to increase public awareness to improve public knowledge of and attitudes toward the effective use of antimicrobials and understanding of antimicrobial resistance [16]. Among the areas targeted for public awareness are school communities [16]. Antibiotics are among the commonly dispensed antimicrobial drugs in Kenya [17]. In a study that used drug resistant index (DRI) as a measure of antibiotic effectiveness where a score of 25% and below was considered indicative of antibiotic resistance under control, Kenya's DRI score of 56.20% highlighted the antibiotic resistance crisis that was attributed to, among other factors, high infection burden and inadequate access to quality water and sanitation services [18]. Despite having identified school communities for public education to raise awareness of proper antimicrobial use, there is limited information regarding teachers' knowledge of and attitudes towards antibiotic use and resistance. This information is important in positioning teachers as

key players in disseminating AMR awareness in schools and implementing effective interventions for the rational use of antibiotics. Therefore, this study sought to determine the following:

i. The level of knowledge of antibiotic use and antibiotic resistance among teachers in Kenya

ii. The extent to which attitudes among teachers in Kenya impact antibiotic use and antibiotic resistance

iii. Predicators of adequate knowledge about and adequate attitudes towards antibiotic use and antibiotic resistance among teachers in Kenya

## Materials and methods

### Study setting and design

This is a cross-sectiocal study to examine knowledge on and attitudes towards antibiotic use and antibiotic resistance among teachers in Kenya. The study was conducted from 1st December 2023 to 31st January 2024. Teachers Service Commission Sub County directors, who are in charge of teacher management in the 271 sub counties in Kenya, were first taken through the study design and objectives by the study authors. The study objectives were then explained to the participants via virtual meetings that were conducted by the Teachers' Service Commission Sub Cunty Directors at the onset of the study period. The participants were required to read and agree to provide informed consent before completing the questionnaire. Participants who did not agree to provide informed consent were not allowed to proceed with the study. The potential risks and benefits of the study were well explained. The respondents' confidentiality was assured by collection of data that were completely anonymous and that no personal identifiable information was collected. The respondents were required to consent to voluntary participation and were free to choose to not participate or end their participation at any time of the study. The participants were free to contact the study authors if they had any questions or comments.

### Inclusion and exclusion criteria

The study sampled and included registered teachers teaching in primary and secondary schools across Kenya and employed by the Teachers Service Commission. Registered but unemployed teachers were not included in the study. Furthermore, only teachers who read and agreed to provide informed consent, and submitted their responses to the online questionnaire, were included in the final study sample.

### Sample size

There are 222, 000 teachers teaching in 32,594 primary schools and 128,000 teachers teaching in 10,482 secondary schools in the Republic of Kenya. The teachers are employed by the Teachers Service Commission of Kenya. Additionally, school communities comprise an estimated 10.4 million learners and 3.9 million students in primary and secondary schools, respectively [19]. The sample size was determined using Raosoft sample size calculator [20] for a population of 350,000 (the number of teachers working in both primary and secondary schools) with a 95% confidence interval and a 5% margin error, the minimum number of the estimated sample was 384. Since this is the first cross-sectional study on AMR involving teachers drawn from primary and secondary schools in Kenya, the sample size was increased by approximately 70% to increase the representative sample size, resulting in a final sample size of

653. In Kenya, there are 271 sub-counties. To ensure a representative sample of the teachers in the country, respondents were required to indicate their schools' sub-county for random proportional distribution of the submitted questionnaires on the basis of the number of teachers in each sub-county.

## Data collection and analysis

A questionnaire that was used in a previous study [21] was slightly modified to reflect respondents' sub counties and sources of antibiotics, and then validated through a test-retest reliability method involving 100 randomly sampled teachers in Kwale County, Kenya. The questionnaire had three sections: The first section sought information on socio-demographic characteristics of the respondents, the source of antibiotics and on whether the respondents had been exposed to educational materials on antibiotic use and resistance. The second section contained 17 questions for assessing respondents' knowledge on antibiotic use and antibiotic resistance whereas the last section included 11 items that assessed respondents' attitudes toward antibiotic use. Online questionnaires in the form of Google forms were shared with the Teachers Service Commission Sub County Directors for distribution to randomly selected teachers in the Republic of Kenya. The questionnaires were coded, and the collected data were entered into an Excel file. IBM SPSS Statistics 27 software was used for all statistical analyses. A $P \leq 0.05$ was considered significant. Categorical data were presented as frequency and percentages. Continuous data were checked for normality and then presented as mean and standard deviation (SD) for normally distributed variables.

## Results

### (A) Demographic characteristics of the respondents

A total of 708 teachers submitted their responses to the online questionnaire; however, 100 questionnaires were excluded from the study. Therefore, 608 completed questionnaires constituted the final sample representing a response rate of 93.1% of the estimated sample size of 653. The respondents included 369 males and 239 females accounting for 60.7% and 39.3% of the respondents respectively. The median age group of the respondents was 40–49 years. Half of the sampled teachers had degree certificates, 36.2% had a certificate or diploma and 13.8% had postgraduate qualifications. Primary and secondary school teachers constituted 56.1% and 43.9% of the respondents respectively. Over 65% of the respondents sourced antibiotics from a health center on prescription while 34.4% purchased antibiotics from private pharmacies without a medical prescription. More than 82% of the sampled teachers had not been exposed to public awareness information on proper antibiotic use and antibiotic resistance. *Table 1* shows a summary of the respondents' demographic characteristics.

### (B) Respondents' knowledge of antibiotic use and antibiotic resistance

A total of 17 questions were used to measure respondents' knowledge of antibiotic use and antibiotic resistance. The antibiotic knowledge score was calculated as the proportion of correct or incorrect responses submitted for every 608 responses per question and the quotient multiplied by 17. The mean knowledge score for antibiotic use and resistance was 9.2 (*SD* = 3.48), while the median score was 8.5. Adequate knowledge scores on antibiotic use and resistance were considered for values ranging from 9–17 whereas inadequate knowledge scores were within the range of 0–8. Only 53.3% of the respondents affirmed to be able to differentiate between viral and bacterial infections whereas over 71% correctly responded that viruses cause the most colds and coughs. However, 86% of the respondents incorrectly responded that

**Table 1. Demographic characteristics of the respondents.**

| Participants' Characteristics | Response options | N = 608 | Percentage |
|---|---|---|---|
| Age | 18–29 | 59 | 9.7 |
| Gender | 30–39 | 119 | 19.6 |
| Marital Status | 40–49 | 158 | 25.9 |
| Level of Education | 50 and above | 272 | 44.7 |
| Type of school | Male | 369 | 60.7 |
| Source of antibiotics | Female | 239 | 39.3 |
| Information about antibiotics | Married | 517 | 85 |
| | Single | 75 | 12.3 |
| | Divorced | 6 | 0.99 |
| | Windowed | 10 | 1.64 |
| | Certificate/Diploma | 220 | 36.2 |
| | Degree | 304 | 50 |
| | Postgraduate | 84 | 13.8 |
| | Primary | 341 | 56.1 |
| | Secondary | 267 | 43.9 |
| | Health center (prescription) | 399 | 65.63 |
| | Pharmacies (non-prescription) | 209 | 34.37 |
| | Yes | 109 | 17.93 |
| | No | 499 | 82.07 |

antibiotics are prescribed for most colds and coughs. More than 76% and 49% of the respondents correctly and incorrectly said that antibiotics kill bacteria and viruses respectively. When asked whether bacteria that live on the skin or in the gut are good for health, 56.25% of the respondents said the bacteria are not good for the skin or for the gut. Additionally, 72.4% of the interviewees incorrectly responded that antibiotics do not kill bacteria that normally live in the skin and in the gut. On antibiotic resistance, over 79% of the sampled teachers correctly indicated that antibiotic resistance means bacteria will not be killed by antibiotics. More than 59% of the respondents correctly answered that antibiotic resistant infections are not easily cured or may not be cured by antibiotics. Over 84% of the sampled teachers correctly indicated that prolonged use of antibiotics can make bacteria resistant, but surprisingly, more than 65% of the respondents incorrectly agreed that if antibiotics are taken less than the prescribed dose, bacteria become less resistant. More than 32% of the interviewees incorrectly responded that prescribed dose and duration of antibiotics can be stopped if symptoms improve. Only 47.3% of the respondents correctly responded that antibiotic resistance can be spread between bacteria whereas 25.8% incorrectly responded that antibiotics have no side effects. *Table 2* is summary of the responses obtained in each of the 17 questions.

## (C) Respondents' attitudes towards antibiotic use and antibiotic resistance

Eleven questions were used to assess respondents' attitudes towards antibiotic use and resistance. An attitude score was determined by calculating the number of appropriate responses to the 11 questions and a mean attitude score of 6.1 (SD = 2.7) was obtained. The median attitude score was 5.4. The good attitude score was within the range of 6–11 whereas the poor attitude scores were within the range of 0–5. The respondents had adequate attitude scores for four questions representing, 36.4% of the 11 questions on attitudes towards antibiotic use and resistance. More than 75% of the respondents incorrectly indicated that they would expect antibiotics to be prescribed by their doctors if they experienced colds, and 70.9% of the respondents incorrectly agreed that they would ask for antibiotic prescriptions to prevent symptoms from worsening when they experienced colds. More than a quarter of the respondents agreed that they would stop taking antibiotics once symptoms improved whereas 25.7% of the sampled teachers admitted to giving family members their prescribed antibiotics wherever the family

**Table 2. Knowledge of antibiotic use and antibiotic resistance among teachers in Kenya.**

| S/No | Question on antibiotic knowledge | Knowledge score | N = 608 (%) | |
|---|---|---|---|---|
| | | | Correct response | Incorrect response |
| 1 | I can differentiate between bacterial and viral infection | 9.05 | 324 (53.3) | 284 (46.7) |
| 2 | Viruses cause most cold and cough | 12.23 | 437 (71.86) | 171 (28.14) |
| 3 | Antibiotics are prescribed for most cold and cough | 2.32 | 83 (13.65) | 525 (86.35) |
| 4 | Antibiotics are effective for most sore throat | 5.14 | 184 (30.26) | 424 (69.74) |
| 5 | Antibiotics can kill bacteria | 13.08 | 468 (76.97) | 140 (23.03) |
| 6 | Antibiotics can kill viruses | 8.67 | 310 (50.98) | 298 (49.02) |
| 7 | Bacteria that live normally on the skin and in the gut, are good for the health | 7.43 | 266 (43.75) | 342 (56.25) |
| 8 | Antibiotics does not kill the bacteria that live normally on the skin and in the gut | 4.69 | 168 (27.63) | 440 (72.37) |
| 9 | Antibiotics are the same as the medications used to relief pain and fever | 10.93 | 391 (64.30) | 217 (35.70) |
| 10 | Antibiotics resistance means that bacteria will not be killed by antibiotics | 13.56 | 485 (79.76) | 123 (20.24) |
| 11 | Infections caused by antibiotic resistant bacteria cannot be easily cured or cannot be cured | 10.12 | 362 (59.54) | 246 (40.46) |
| 12 | If antibiotics are taken for long period of time, bacteria become resistant to antibiotics | 14.32 | 512 (84.21) | 96 (15.79) |
| 13 | If antibiotics are taken less than the prescribed dose, bacteria become less resistant to antibiotics | 5.90 | 211 (34.70) | 397 (65.30) |
| 14 | If twice the prescribed dosage of antibiotic is taken, the effects of antibiotics are more rapid | 7.19 | 257 (42.27) | 351 (57.73) |
| 15 | The prescribed dose and duration of antibiotics can be terminated if the symptoms improve | 11.52 | 412 (67.76) | 196 (32.24) |
| 16 | Antibiotic resistance can spread between bacteria | 8.05 | 288 (47.37) | 320 (52.63) |
| 17 | Antibiotics have no side effects | 12.61 | 451 (74.18) | 157 (25.82) |

members fell sick. Slightly more than half of the respondents agreed that antibiotics should be kept at home for emergencies. However, more than 97% of the sampled teachers correctly agreed to take antibiotics according to the instructions on the label. Table 3 shows a summary of the attitude scores and correct and incorrect responses to questions concerning attitudes towards antibiotic use and resistance.

## (D) Associations between groups and their level of knowledge and attitudes towards antibiotic use and antibiotic resistance

A chi-test was used to determine the statistical associations between groups (age, sex, marital status, level of education and type of school a teacher taught) and their level of knowledge on antibiotic use and resistance as well as the associations between the groups and their attitudes

**Table 3. Attitudes towards antibiotic use and resistance among Kenyan teachers.**

| S/No | Question | Attitude score | N = 608 (%) | |
|---|---|---|---|---|
| | | | Correct | Incorrect |
| 1 | I expect antibiotics to be prescribed by my doctor if I suffer from common cold symptoms | 2.66 | 147 (24.17) | 461(75.83) |
| 2 | If I catch a cold, I ask for antibiotic prescription to prevent my symptoms from getting worse | 3.20 | 177 (29.11) | 431 (70.89) |
| 3 | I believe that antibiotics cure my cold faster | 2.99 | 165 (27.13) | 443 (72.87) |
| 4 | I take left-over antibiotics when I have flu or other symptoms | 9.23 | 510 (83.88) | 98 (16.12) |
| 5 | I stop taking the prescribed antibiotics once I get better | 8.10 | 453 (74.5) | 155 (25.5) |
| 6 | I prefer a shot (injection) to an oral medication if antibiotics are needed | 4.88 | 270 (44.4) | 338 (55.60) |
| 7 | I check to see if the antibiotics are included within the prescribed cold medicine | 5.57 | 308 (50.66) | 300 (49.34) |
| 8 | I know which medication is an antibiotic when I take cold medicines | 5.79 | 320 (52.63) | 288 (47.37) |
| 9 | If my family member is sick, I usually give my prescribed antibiotic to them | 8.18 | 452 (74.34) | 156 (25.66) |
| 10 | I normally keep antibiotic stock at home in case of an emergency | 5.28 | 292 (48.03) | 316 (51.97) |
| 11 | I take antibiotics according to the instructions on the label | 10.69 | 591 (97.20) | 17 (2.80) |

towards antibiotic use and antibiotic resistance. There was no significant ($P > 0.05$) association between any of the groups and their level of knowledge of antibiotic use and resistance or with their attitudes towards antibiotic use and antibiotic resistance.

## (E) Associations between knowledge of and attitudes toward antibiotic use and antibiotic resistance

To evaluate the associations between knowledge of and attitude towards antibiotic use and resistance, a *Phi* test was conducted and a positive association (*Phi* = 0.22, $P = 0.05$) was obtained. A binary logistic regression was conducted to determine predicators of adequate knowledge of antibiotic use and antibiotic resistance as well as predicators of good attitudes towards antibiotic use and resistance. A total of 392 respondents had adequate knowledge of antibiotic use and resistance whereas 216 had inadequate knowledge of antibiotic use and antibiotic resistance. The good predicators of adequate knowledge of antibiotic use and resistance included respondents aged 40–49 years (OR = 1.959, CI = 1.057–3.631, $P = 0.033$) who were 1.959 times more likely to have adequate knowledge of antibiotic use and antibiotic resistance and secondary school teachers (OR = 7.437, CI = 6.688–8.199, $P = 0.043$) who were 7.437 times more likely to have adequate knowledge of antibiotic use and antibiotic resistance. However, respondents who were single (OR = 0.276, CI = 0.56–1.367), divorced (OR = 0.833, CI = 0.142–4.877), degree holders (OR = 0.759, CI = 0.379–1.520) or teaching in primary school (OR = 0.757, CI = 0.401–1.334) were less likely to have adequate knowledge of antibiotic use and antibiotic resistance. A total of 256 respondents had scores for adequate attitude towards antibiotic use and resistance whereas 356 had inadequate scores and, therefore, had poor attitudes towards antibiotic use and resistance. Predicators of good attitude towards antibiotic use and resistance were obtained for the sampled teachers aged 30–39 (OR = 2.467, CI = 0.999–6.093, $P = 0.05$), and 40–49 (OR = 1.927, CI = 1.021–3.635, $P = 0.043$), who were 2.467 and 1.927 times more likely to have adequate attitude towards antibiotic use and resistance respectively. However, respondents who were single (OR = 0.666, CI = 0.70–2.610), divorced (OR = 0.659, CI = 0.142–3.051), having degree (OR = 0.559, CI = 0.246–1.274) and teaching in a primary school (OR = 0.88, CI = 0.488–1.586) were less likely to have an adequate attitude towards antibiotic use and resistance.

## Discussion

This is the first study to be conducted to comprehensively demonstrate knowledge and attitudes towards antibiotic use and antibiotic resistance among Kenyan teachers. Therefore, the present findings provide baseline quantitative data patterns on knowledge and attitudes towards antibiotic use and antibiotic resistance among Kenyan teachers. The Kenya Antimicrobial Interagency Stewardship Committees have identified school communities for public education campaigns to improve awareness of antimicrobial use and antimicrobial resistance as a strategy to ensure rational use of antimicrobials and to reduce the increasing threat posed by antimicrobial resistant infections [15]. However, there is a dearth of data on public awareness of the proper use of antimicrobials and antimicrobial resistance among Kenyan school teachers. The present study sought to identify knowledge and attitudes gaps among the Kenyan school teachers in primary and secondary schools across the country to provide data to Kenya Antimicrobial Interagency Stewardship Committees and other interested public health stakeholders to formulate effective policies for mitigating the gaps in the effective use of teachers in disseminating information for rational antimicrobial use in school communities. In this study, the response rate was 93.1% with 608 of the sampled teachers submitting completed online questionnaires. Among the respondents, 82% indicated that they had not been exposed

to public information on awareness of proper antibiotic use and antibiotic resistance. The low number of teachers who had been exposed to information on antibiotic use and resistance may explain the knowledge gaps identified in the current study. For example, over 86% of the sampled teachers incorrectly responded that antibiotics are prescribed for most colds and coughs whereas 72% indicated that antibiotics do not kill useful bacteria found in the gut and skin. More than one-third of the respondents agreed that antibiotics should be stopped when symptoms improve. These knowledge gaps concerning antibiotic use are misconceptions that put patients at risk of relapse, especially those with resistant disease-causing bacteria [22]. Studies related to knowledge of and attitudes towards antimicrobial use and resistance among non-health professionals within East Africa are scarce. In Uganda, for example, a cross-sectional study on knowledge, perceptions and practices related to antimicrobial resistance in humans and animals among Wakiso district inhabitants revealed that 36.4% of the respondents had not heard of AMR [23]. Additionally, 70.8% of the participants reported that resistant microorganisms cause infections that are difficult to treat [23]. This finding is comparable to that reported in this study, where 79.76% of the respondents reported that antibiotic-resistant bacteria are not killed by antibiotics. In a meta-analysis examining the prevalence of knowledge, attitudes and practices regarding antimicrobial resistance in Africa, the pooled knowledge of antimicrobial use and resistance was reported to be 55.33%, while the pooled positive attitude score was 46. 93% [24]. This study reported comparable scores of 54.12% and 55.45% for knowledge of and attitudes towards antibiotic use and resistance, respectively. Inadequate dosing, incomplete courses and the sourcing of antibiotics from private pharmacies without medical prescriptions contributes to the emergence and spread of antimicrobial resistance, which is a current public health problem in Kenya [25]. While Kenya has regulations prohibiting the sale of antibiotics without medical prescription [26], more than 34% of respondents sourced the drugs from private pharmacies without prescription, therefore, in addition to creating public awareness of the rational use of antibiotics, there is a need to enforce the regulations. The prevalence of negative attitudes among the sampled teachers highlights the need for public education about the effectiveness and resistance of antibiotics. Public education campaigns should endeavor to provide information on proper and practical ways to change teachers' attitudes and behaviours to ensure proper antibiotic use in school communities. Studies indicate that increasing communities' knowledge of antibiotic use and resistance without imparting appropriate attitudes can result in increased incidences of antibiotic misuse [26,27]. In the present study, there were no significant associations between the level of the sampled teachers' knowledge of and attitudes towards antibiotic use and resistance and background characteristics (age, gender, marital status, and level of education). This implies that education campaigns targeting teachers in school communities in Kenya should consider the development of comprehensive and multifaceted interventions, such as the use of campaigns to alter attitudes towards self-medication and the enforcement of stricter regulations aimed at diminishing the availability of antibiotics without prescription. A positive, albeit weak, correlation was observed in this study between the respondents' knowledge and attitudes. Therefore, efforts focused on reaching identified groups of teachers who have low knowledge and negative attitudes towards antibiotic use and resistance should inform future public campaigns in schools to improve knowledge and change attitudes and behaviours towards antibiotic use and resistance. In the present study, the sampled teachers who were single, divorced, having a degree, and teaching in a primary school were identified as a group that was less likely to have adequate knowledge of antibiotics use as well as inadequate attitudes towards antibiotic use and resistance.

## Limitations of the study

This study used a self-administered questionnaire and, therefore, relied upon information given by the sampled teachers; this kind of approach is prone to recall bias. It is possible that some respondents overreported desirable behaviours and underreported socially undesirable behaviours. However, only questionnaires that were fully completed were included in the final analysis. Furthermore, the respondents were assured of confidentiality. These steps were taken to minimize over and underreporting of information. This is a cross-sectional study whose data characterize a single time instance and, therefore, does not consider changes in respondents' attitude and knowledge in relation to antibiotic use and resistance over time. However, this study is useful for evaluating and improving knowledge of and attitudes towards antibiotic use and resistance in Kenyan school communities by targeting teachers.

## Conclusions

The findings of this study should motivate teacher training colleges and universities to consider introducing a common unit in antimicrobial use and resistance among teacher trainees to ensure that graduating teachers have adequate knowledge and attitudes on antimicrobial use. This will help school communities and the general public to benefit from the teachers' knowledge of antibiotic use upon graduation.

## Supporting information

**S1 Questionnaire.**
(DOCX)

## Acknowledgments

We gratefully acknowledge the study respondents without whom this survey would not have been possible. We remain indebted to all the Teachers' Service Commission Sub County Directors, who were instrumental in ensuring the online distribution of the online questionnaires to the respondents. We extend our gratitude to the anonymous reviewers and the academic editor, Ali Haider Mohammed, for their constructive comments and suggestions.

## Author Contributions

**Conceptualization:** Patrick M. Mutua, Joshua. Mutiso, Michael M. Gicheru.

**Data curation:** Patrick M. Mutua, Joshua. Mutiso.

**Formal analysis:** Patrick M. Mutua, Joshua. Mutiso, Michael M. Gicheru.

**Methodology:** Patrick M. Mutua, Michael M. Gicheru.

**Project administration:** Patrick M. Mutua, Joshua. Mutiso.

**Resources:** Patrick M. Mutua.

**Software:** Patrick M. Mutua, Michael M. Gicheru.

**Supervision:** Michael M. Gicheru.

**Validation:** Michael M. Gicheru.

**Writing – original draft:** Patrick M. Mutua, Joshua. Mutiso, Michael M. Gicheru.

**Writing – review & editing:** Patrick M. Mutua, Joshua. Mutiso, Michael M. Gicheru.

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
