## [Decision Letter · Decision Letter 0]

23 Sep 2024

PONE-D-24-18988A Survey of Knowledge and Attitude Towards Antibiotic use and Resistance among the Teachers in the Republic of Kenya: Implications for Using Teachers in Raising Public Awareness on Rational Antibiotic use in School CommunitiesPLOS ONE

Dear Dr. Mutua,

Thank you for submitting your manuscript to PLOS ONE. After careful consideration, we feel that it has merit but does not fully meet PLOS ONE’s publication criteria as it currently stands. Therefore, we invite you to submit a revised version of the manuscript that addresses the points raised during the review process.

Kindly find below my comments as well as the reviewers comments .

We look forward to receiving your revised manuscript.

Kind regards,

Ali Haider Mohammed

Academic Editor

PLOS ONE

Journal Requirements:

3. In the online submission form, you indicated that the data underlying the results presented in the study are available from the corresponding author upon request

Additional Editor Comments:

Methodology:

Clarification of Sample Selection: The authors should clarify the selection process for the teachers involved in the survey, particularly how they ensured a representative sample across various regions and school types within Kenya.

Detailed Statistical Analysis: While the statistical tools used (e.g., IBM SPSS) are mentioned, the manuscript would benefit from a more detailed explanation of the statistical methods applied, especially the models used in logistic regression.

Results:

Presentation of Data: Some tables are dense and could be simplified or better explained in the text to help readers understand the data without constantly referring to the tables.

Comparative Analysis: It would enhance the manuscript if the authors compared their findings with similar studies in other regions or countries, discussing any notable similarities or differences.

Discussion:

Broader Context: The discussion should integrate more about how these findings align or contrast with global trends in antimicrobial resistance education, especially among non-health professionals.

Implications for Policy and Practice: While the authors touch upon the implications of their findings, they could expand on specific policy recommendations or educational strategies tailored for teachers.

Reviewers' comments:

Reviewer's Responses to Questions

**Comments to the Author**

1. Is the manuscript technically sound, and do the data support the conclusions?

Reviewer #1: Yes

2. Has the statistical analysis been performed appropriately and rigorously? 

Reviewer #1: Yes

3. Have the authors made all data underlying the findings in their manuscript fully available?

Reviewer #1: Yes

4. Is the manuscript presented in an intelligible fashion and written in standard English?

Reviewer #1: No

5. Review Comments to the Author

Reviewer #1: Greetings, thank you for sharing your research outcome.

Kindly double check on some grammar and typing errors in the manuscript.

Eg: Google was typed as “goggle” in line 112, and the word “Google” is usually not capitalized in the manuscript.

Kindly double check on the reference too.

Eg: Reference 15 and 24 appear to be duplicated.

There are multiple different referencing styles used in Reference List, suggest to standardize based on PLOS journal’s guideline.

In a Knowledge and Attitude study, the questions under “Knowledge” section should ideally be used to identify knowledge gap and misconception.

Under Table 2 (Knowledge section), question no.1 asked “I can differentiate between bacterial and viral infection”. In my humble opinion, this question seems to ask the respondents’ perception/reflection regarding their own knowledge. There is no “incorrect” or “correct” response to be judged in such question, and perhaps it might not be suitable to be placed under “Knowledge” section. Can you kindly clarify?

In Line 127, the authors mentioned that the study instrument has previously been used in a different study (a study from Nigeria - Reference 21), and for the purpose of this research, the authors had validated it with 100 randomly sampled teachers in Kenya. However, the actual questionnaire used in this manuscript is actually modified and not identical with the study from Nigeria (Reference 21). Suggest to clarify this in the research methodology too.

Thank you.

6. PLOS authors have the option to publish the peer review history of their article (what does this mean?). If published, this will include your full peer review and any attached files.

Reviewer #1: No

---

## [Author Response · Author response to Decision Letter 0]

26 Sep 2024

From: Dr Patrick M. Mutua To: The Editor

 Department of Public Health, The PLoS ONE Journal

 School of Health and Human Sciences, 

 Pwani University P.O Box 195-80108, 

 Kilifi, Kenya

 26th September, 2024

 Email: patmbuvi@gmail.com

Dear Sir/Madam

RE: REBUTTAL ON THE REVISED POINTS IN MY MANUSCRIPT PONE-D-24-18988 

Thank you so much for you email dated 24th September, 2024 requesting me to carry out revisions on our manuscript referenced above. My corrections are as follows:

To the academic Editor:

(1) I have carefully re-written references in line with PLOS one guidelines as advised.

(2) I have added more information in methods section indicating that consent was informed. Additionally, the draft of the questionnaire that was used to design the online questionnaire starts with consent information that requested participants to read and understand the informed consent and either agree on the statement before responding to the online questionnaire or disagree to the consent and therefore decline to proceed with the study. Kindly refer to the questionnaire submitted together with the revised manuscript

(3) In line with meeting the PLOS One journal requirement on availing data used in a study, I have submitted the questionnaire used to collect data as part of the attachment of the main manuscript. Kindly refer to the upload revised manuscript

(4) On sample selection, Under the Sample size section, I have included a statement on how we ensured data on sampled teachers was representative of the teachers by sub county and by school. The statement reads “In Kenya, there are 271 sub-counties. To ensure a representative sample of the teachers in the country, respondents were required to indicate their schools’ sub-county for random proportional distribution of the submitted questionnaires based on the number of teachers in each sub-county.” Additionally, the study involved all 271 Teachers Service Commission Sub County directors who distributed the online questionnaire to the teachers in the sub counties, providing opportunity for all teachers in all the sub counties to respond to the questionnaire.

(5) Under Methods and in the abstract, I have included Binary Logistic Regression as the model that was used to perform statistical analysis to determine the predicators of adequate knowledge and attitude toward antibiotic use and resistance.

(6) Under discussion section, I have included related studies in East Africa and Africa that included non-professionals in assessing knowledge for and attitude toward antibiotic use and resistance. I have further demonstrated how the findings of the related studies compare with the current study

(7) On Implications for policy and practice: In the conclusions section, the study recommends that policy makers in Kenya to use the study findings to plan comprehensive public education campaigns to mitigate gaps identified by the study so as to improve teachers’ knowledge on and attitude for antibiotic use. Further, the study recommends teacher training in colleges in universities to consider offering a course on AMR to teacher trainees. This will equip graduating teachers to have adequate knowledge and attitude on AMR and benefit school communities and the general public 

To the Reviewers:

(1) I have carefully effected corrections to ensure all grammatical errors are corrected and I have highlighted the corrections for your perusal. The word goggle has been corrected to read google in line 137. 

(2) Duplication of reference 15 and 24 has been corrected by deleting reference 24. 

(3) References have been standardized based on PLOS journal guidelines

(4) On whether the question “I can differentiate between bacterial infections from Viral infections” is a knowledge or a perception question, we strongly feel the question is a knowledge one because it is assessing factual understanding. For one to respond to the question, they must have scientific knowledge to distinguish between bacterial and Viral infections based on their characteristics, causes, or symptoms. We, therefore, kindly request that the question be retained as it is in Table 2.

(5) We confirm that the questionnaire that we used in the study was previously used in another study in Nigeria. However, we carried out slight modifications to include a question on sub county of the respondent so as to ensure a representative sample of teachers in Kenya was well captured. We also removed a few items that sought history of use of antibiotics by respondents.

In conclusion, I believe that I have responded to all the corrections I have been requested to ensure are done and that my revised manuscript now meets all the PLoS ONE requirements for publication. I look forward to getting published in your esteemed journal. T

Regards

Dr Patrick M. Mutua

---

## [Decision Letter · Decision Letter 1]

15 Nov 2024

PONE-D-24-18988R1A Survey of Knowledge and Attitude Towards Antibiotic use and Resistance among the Teachers in the Republic of Kenya: Implications for Using Teachers in Raising Public Awareness on Rational Antibiotic use in School CommunitiesPLOS ONE

Dear Dr. Mutua,

Thank you for submitting your manuscript to PLOS ONE. After careful consideration, we feel that it has merit but does not fully meet PLOS ONE’s publication criteria as it currently stands. Therefore, we invite you to submit a revised version of the manuscript that addresses the points raised during the review process.  Reviewers highlighted minor issues that need to be addressed. 

We look forward to receiving your revised manuscript.

Kind regards,

Ali Haider Mohammed

Academic Editor

PLOS ONE

Journal Requirements:

Reviewers' comments:

Reviewer's Responses to Questions

**Comments to the Author**

1. If the authors have adequately addressed your comments raised in a previous round of review and you feel that this manuscript is now acceptable for publication, you may indicate that here to bypass the “Comments to the Author” section, enter your conflict of interest statement in the “Confidential to Editor” section, and submit your "Accept" recommendation.

Reviewer #2: (No Response)

2. Is the manuscript technically sound, and do the data support the conclusions?

Reviewer #2: Yes

3. Has the statistical analysis been performed appropriately and rigorously? 

Reviewer #2: Yes

4. Have the authors made all data underlying the findings in their manuscript fully available?

Reviewer #2: Yes

5. Is the manuscript presented in an intelligible fashion and written in standard English?

Reviewer #2: Yes

6. Review Comments to the Author

Reviewer #2: Couple of minor comments

ABSTRACT

- i don't like "dearth of information" - change to Limited studies have investigated teachers

- in the abstract, it is not clear who "teachers" are = can you specify it's primary and secondary school teachers. So there are no university lecturers in your sample?

- the phrase "the average knowledge score on antibiotic use" is too vague in the abstract, please be specific. Knowledge can include many things. Either pick one big result or explain what you mean by knowledge. The abstract should be able to be read alone without having to go to the main text

- need to add summary of number of respondents and response rate in abstract

MAIN TEXT

- I think you mention 5 times that "this is the first study to conduct knowledge is kenyan teachers" - i don't think you need to keep mentioning this so many times, reduce it and only mention in the discussion and maybe one more time in the intro

- you did some analysis in the results regarding demotraphics of respondents and their attitude to AMR. can you discuss this in your discussion?

- the one odd part i see in your study is that you investigated knowledge of primary and secondary teachers rather than tertiary. do you expect primary school and secondary teachers to teach AMR to their students? If you see a case for this foundation skill all the way back in primary school, please explain in discussion why.

- there's too much information in your conclusion, you have rewritten most of your introduction again, remove first 4 sentences and start with the findings of this study..

- the findings of this study provoke - change to motivate??

7. PLOS authors have the option to publish the peer review history of their article (what does this mean?). If published, this will include your full peer review and any attached files.

Reviewer #2: No

---

## [Author Response · Author response to Decision Letter 1]

16 Nov 2024

From: Dr Patrick M. Mutua To: The Editor

Department of Public Health, The PLoS ONE Journal

School of Health and Human Sciences, 

Pwani University P.O Box 195-80108, 

Kilifi, Kenya

16th November, 2024

Email: patmbuvi@gmail.com

Dear Sir/Madam

RE: REBUTTAL ON THE REVISED POINTS IN MY MANUSCRIPT PONE-D-24-18988R1 

Thank you so much for your email dated 15th November, 2024 requesting me to carry out revisions on our manuscript referenced above. My corrections are as follows:

To the academic Editor:

(1) I have carefully checked all my references and I confirm that none of the references quoted in our manuscript is retracted 

To the Reviewers:

Corrections in the abstract 

(1) I have deleted the sentence “dearth of information…” and replaced this with “limited studies have investigated teachers’ knowledge of and attitudes towards antimicrobial use” 

(2) In the abstract, I have clearly indicated the study focused on primary and secondary school teachers. 

(3) With regard to use of knowledge, we have clearly indicated the knowledge score of teachers of antimicrobial resistance to be 9.2 out of 17. 

(4) In the abstract, I have included the response rate of the participants as advised 

Corrections in main text

(1) I have reduced the number of times I have mentioned “This is the first study to” and only mentioned this in the discussion as advised

(2) In our manuscript, we have quoted a related study done in Ghana that used storytelling to positively improve knowledge of antibiotics and antimicrobial resistance among children attending primary schools. This study involved primary school teachers. Therefore, it is possible for primary and secondary school teachers to teach about antimicrobial resistance using a variety of learner centered approaches as is the case with the Ghana study

(3) I have deleted the first four sentences in my conclusion as advised and the revised section starts with the findings

(4) I have used the word to MOTIVATE rather than to provoke…in the conclusions section with regard to using the findings to MOTIVATE teacher training colleges and universities to consider a course of antimicrobial resistance for teacher trainees 

In conclusion, I believe that I have responded to all the corrections I have been requested to ensure are done and that my revised manuscript now meets all the PLoS ONE requirements for publication. I look forward to getting published in your esteemed journal. T

Regards

Dr Patrick M. Mutua

---

## [Editor Report · Decision Letter 2]

6 Dec 2024

A Survey of Knowledge and Attitudes Towards Antibiotic Use and Resistance among Teachers in the Republic of Kenya: Implications for Using Teachers in Raising Public Awareness of Rational Antibiotic Use in School Communities

PONE-D-24-18988R2

Dear Dr. Patrick ,

Taking into consideration that you have significantly addressed the reviewers' comments and improved the quality of the paper, we’re pleased to inform you that your manuscript has been judged scientifically suitable for publication and will be formally accepted for publication once it meets all outstanding technical requirements.

Kind regards,

Ali Haider Mohammed

Academic Editor

PLOS ONE

Additional Editor Comments (optional):

All comments were addressed optimally.
---

## [Editor Report · Acceptance letter]

10 Dec 2024

PONE-D-24-18988R2 

PLOS ONE

Dear Dr. Mutua, 

I'm pleased to inform you that your manuscript has been deemed suitable for publication in PLOS ONE. Congratulations! Your manuscript is now being handed over to our production team.

Kind regards, 

on behalf of

Dr. Ali Haider Mohammed 

Academic Editor

PLOS ONE